# Pancreatic Fibroblast Heterogeneity: From Development to Cancer

**DOI:** 10.3390/cells9112464

**Published:** 2020-11-12

**Authors:** Paloma E. Garcia, Michael K. Scales, Benjamin L. Allen, Marina Pasca di Magliano

**Affiliations:** 1Program in Molecular and Cellular Pathology, University of Michigan Medical School, University of Michigan, Ann Arbor, MI 48105, USA; palomag@umich.edu; 2Department of Cell and Developmental Biology, University of Michigan Medical School, University of Michigan, Ann Arbor, MI 48109, USA; mkscales@umich.edu (M.K.S.); benallen@umich.edu (B.L.A.); 3Rogel Cancer Center, Michigan Medicine, University of Michigan, Ann Arbor, MI 48109, USA; 4Department of Surgery, Michigan Medicine, University of Michigan, Ann Arbor, MI 48109, USA

**Keywords:** fibroblasts, pancreas, cancer-associated fibroblasts, pancreas development, pancreatic cancer, tumor microenvironment, hedgehog, transforming growth factor Beta

## Abstract

Pancreatic ductal adenocarcinoma (PDA) is characterized by an extensive fibroinflammatory microenvironment that accumulates from the onset of disease progression. Cancer-associated fibroblasts (CAFs) are a prominent cellular component of the stroma, but their role during carcinogenesis remains controversial, with both tumor-supporting and tumor-restraining functions reported in different studies. One explanation for these contradictory findings is the heterogeneous nature of the fibroblast populations, and the different roles each subset might play in carcinogenesis. Here, we review the current literature on the origin and function of pancreatic fibroblasts, from the developing organ to the healthy adult pancreas, and throughout the initiation and progression of PDA. We also discuss clinical approaches to targeting fibroblasts in PDA.

## 1. Introduction

Fibroblasts are mesenchymal cells with essential roles throughout embryonic development and adult organ function. Activated fibroblasts, also known as myofibroblasts, temporarily expand to aid in wound repair, while their numbers decline during tissue remodeling [1,2]. When this remodeling fails to occur in the context of disease, the persistence of fibrotic tissue can impair normal tissue function [3].

Certain pancreatic diseases, such as chronic pancreatitis and pancreatic cancer, are characterized by dense fibrosis throughout the organ. In pancreatic ductal adenocarcinoma (PDA) this fibrosis is accompanied by infiltration of inflammatory cells, giving rise to a fibroinflammatory reaction commonly referred to as stroma. Interestingly, this stroma can make up the majority of the tumor area in PDA [4]. The functional role of pancreatic fibroblasts in pancreatic diseases has been the focus of considerable research over the past 20 years. Despite these extensive efforts, the roles of fibroblasts still remain controversial. Numerous studies have identified tumor-supporting roles for cancer-associated fibroblasts (CAFs), such as promoting immunosuppression [5,6], supporting tumor metabolism [7], and facilitating chemoresistance [8,9,10,11]. However, recent data has also identified tumor-suppressive functions within the stroma [12,13,14,15]. These seemingly contradictory findings suggest that CAFs are more heterogeneous than previously thought. To investigate this possibility, researchers are currently utilizing a combination of animal models, human samples, and novel bioinformatic techniques to study the tumor microenvironment (TME) in greater depth. 

Genetically engineered mouse models have been used extensively to study the in vivo complexity of the pancreatic TME. Several mouse models are based on pancreas-specific expression of common PDA mutations, and faithfully re-capitulate the progression and characteristics of human disease. An oncogenic mutation of the *KRAS* gene is almost universally present in PDA [16,17,18]. Mice that express oncogenic *Kras* (K: *Kras^LSL-G12D^*) in the pancreatic epithelium upon Cre recombination (C: *Pdx1-Cre* or *Ptf1a-Cre*) are predisposed to forming pre-cancerous lesions [19,20], and this process can be synchronized and accelerated by inducing pancreatitis [21,22,23]. These “KC” mice are a useful tool for studying the early stages of pancreatic neoplasia. To study metastatic pancreatic cancer in vivo, KC mice have been crossed with loss-of-function tumor suppressor lines, such as mutant *p53* (KPC) [24] and *p16-Ink4a/p19-Arf* (KIC or KPP) [20]. In addition to these Cre-driven models of PDA, researchers have recently generated a FlpO-driven mouse model (KF, K: *Kras^FSF-G12D^*; F: *Pdx1-FlpO* or *Ptf1a-FlpO*) [25,26,27,28]. These models use the FlpO recombinase to activate mutant *Kras* expression in the epithelium, and can then be crossed with other cell type-targeting Cre lines, enabling independent genetic manipulation of multiple cell types within the TME. This is of particular interest to the study of stromal cells in PDA, whose origin, fate, and function has been poorly understood. This combined genetic system allows for cell-type specific lineage tracing, in which a Cre-expressing cell lineage can be identified throughout different stages of PDA progression. Further, this approach can also be used to ablate a specific cell lineage by introducing a Cre-inducible allele expressing the Diphtheria toxin receptor (DTR), rendering the cells susceptible to depletion via diphtheria toxin (DT) administration [29]. Together, the combination of disease modeling with a diverse genetic tool kit makes the mouse a useful system for investigating the role of fibroblasts in pancreatic disease. 

Mouse models become increasingly powerful when paired with large-scale genomic, proteomic, and transcriptomic analysis. Bioinformatic techniques have enabled researchers to characterize distinct cell populations within the TME to a much greater depth than previously possible. In particular, single-cell RNA sequencing has been used to identify distinct transcriptional cell subpopulations in breast [30], colon [31], head and neck [32], and lung cancer [33]. Now, similar techniques are being used to characterize different cell types in the pancreas at different stages of development and disease [34,35,36,37,38]. By analyzing fibroblast populations in both human patients and mouse models, researchers are starting to identify novel patterns of fibroblast heterogeneity. 

In this review, we discuss the growing body of research describing fibroblast heterogeneity in the pancreas. We explore the different populations that have been identified in the embryonic, adult, and diseased pancreas, and present the current challenges facing the field.

## 2. Mesenchyme Function and Heterogeneity during Pancreas Development

During embryogenesis, the pancreatic buds emerge from the gut endoderm, and receive key signaling cues from the lateral plate mesoderm-derived mesenchyme (for review see [39]). The essential role of the mesenchyme in pancreas development was first hypothesized in the 1960s, when pancreas cultures lacking mesenchyme failed to form primitive acinar structures [40,41]. Almost 50 years later, this idea was tested in vivo by conditionally depleting the developing pancreatic mesenchyme. Researchers utilized an *Nkx3.2-Cre* mouse line in combination with a Cre-dependent DTR to ablate a broad mesenchymal population in the developing pancreas, leading to a severe reduction in pancreas growth [42]. Thus, the mesenchyme plays an essential role during pancreas development. More specifically, proper pancreas development requires communication between the mesenchyme and the pancreatic epithelium through an array of inter-connected signaling pathways. 

In the earliest stages of pancreas development, secreted signals from the mesoderm must signal to the developing endoderm to direct pancreas specification. Two key signals include fibroblast growth factor (FGF) and retinoic acid (RA). Low levels of notochord-derived FGF2 induce pancreas-specific gene expression in the foregut endoderm [43]. In addition, FGF10 secreted by the pancreatic mesenchyme plays a key role in early pancreatic development. *Fgf10* mutant mouse embryos exhibit a dramatic reduction in both endocrine and exocrine pancreatic cell types, due to a failure in epithelial progenitor cell proliferation [44]. Similarly, mesoderm-derived RA is required for normal pancreas lineage specification, as RA depletion impairs the foregut endoderm’s ability to commit to a pancreas fate in zebrafish and mice [45,46,47]. Together, these data highlight the importance of mesenchyme-derived signals during epithelial patterning and establishment of the pancreatic lineage. 

In addition to mesenchyme-derived signals acting on epithelium, proper pancreas development also relies on signals that act on the mesenchyme. For example, activation of bone morphogenic protein (BMP) signaling in the mesenchyme is necessary for normal pancreas morphogenesis. In pancreas explants from both chick and mouse embryos, disrupted mesenchymal BMP signaling led to impaired epithelial development, including abnormal branching and an aberrant relative increase in endocrine cells [48]. Inhibiting epithelial BMP in these explants had no reported phenotype, indicating that this effect was due specifically to the role of BMP signaling in the mesenchyme. The absence of endodermal BMP signaling is in fact necessary during organ differentiation to diverge pancreas from liver, as BMP activation in this region of the endoderm during this critical window promotes liver specification at the expense of pancreas identity [49,50,51,52] Thus, careful spatial and temporal regulation of BMP activation is essential for proper pancreas development. 

This concept of carefully coordinated pathway regulation is a common theme in pancreas development. Beyond BMP signaling, the repression of epithelial sonic hedgehog (*Shh*) expression is necessary for proper pancreas lineage specification [43,53,54]. Ectopic *Shh* expression in the pancreatic epithelium drives intestinal differentiation at the expense of pancreas [53], while antibody-mediated inhibition of SHH is sufficient to induce the expression of pancreas differentiation markers in endoderm explants [43]. Further, mesenchyme-specific activation of the Hedgehog (HH) pathway restricts epithelial growth, while stimulating mesenchymal hyperplasia [55]. Coordination of HH-limiting mechanisms in both the epithelium and the mesenchyme is therefore crucial for balanced pancreas morphogenesis. 

Following pancreatic lineage specification, there is a continuing requirement for epithelial-mesenchymal cross-talk for growth and maturation of the pancreas. Mesenchymal *Hox6* plays a key role in this process, as loss of *Hox6* paralogs in the mesenchyme impairs endocrine cell differentiation [56]. Epithelial and mesenchymal Wnt signaling is also necessary for proper pancreas development, as loss of Wnt pathway activation in either compartment reduces pancreas growth [42,57,58]. Conditional epithelial loss of β-catenin, a key transcriptional mediator of Wnt signaling, leads to a dramatic loss of acinar tissue [57,58], whereas conditional deletion in the mesenchyme reduced both acinar and β-cell mass [42]. Although some level of Wnt activation is necessary for pancreas development, excessive Wnt can disrupt morphogenesis. However, the exact effect is highly dependent on developmental stage. When β-catenin is stabilized early in pancreas development (E11.5 and earlier), there is a significant loss of epithelial mass [59]. This hypoplasia is preceded by a loss of mesenchymal FGF10 and epithelial progenitor cells. Conversely, when β-catenin is stabilized later in development (after E12.5), the pancreas undergoes excessive postnatal growth [59]. These data indicate that Wnt-mediated cross-talk is initially crucial to maintain the instructive signals needed for pancreas development, but the role of this cross-talk changes dramatically as development proceeds. Interestingly, Wnt signaling also demonstrates how a single signaling pathway can direct epithelial-to-mesenchymal signaling as well as mesenchymal-to-epithelial signaling, highlighting the dynamic nature of intercellular cross-talk in the developing pancreas. 

Historically, the pancreatic mesenchyme has been viewed as homogenous, comprised of cells all equally competent to respond to different developmental cues. However, recent data indicate that specific mesenchymal subpopulations have enriched activity for particular signaling pathways [34], suggesting a previously under-appreciated level of heterogeneity. Research efforts in the past few years have sought to characterize this mesenchymal heterogeneity in greater detail. For example, a *Nkx3.2*-expressing mesenchymal lineage has been identified in the developing pancreas and plays key roles in both the embryonic and post-natal organ. As previously mentioned, depletion of this population embryonically via DT severely disrupted pancreas morphogenesis [42]. These mesenchymal cells give rise to pericytes in the adult pancreas [60], and depleting these cells in the adult via DT administration reduces β-cell function, leading to disrupted glucose regulation [61]. Proteomic analysis of lineage-traced *Nkx3.2*-expressing cells revealed that this population undergoes dramatic changes in protein expression from embryonic to postnatal stages, and expresses multiple factors that can promote pancreatic progenitor cell differentiation [62]. These data demonstrate that individual lineages of mesenchymal cells are dynamic, and can play unique roles at different stages within the same tissue.

Novel technologies have expanded our concept of heterogeneity within the developing pancreatic mesenchyme by identifying distinct groups of fibroblasts based on transcriptional profile. Single cell RNA sequencing of the pancreatic mesenchyme at different stages of embryonic development revealed seventeen transcriptionally distinct clusters [34]. While some of these cell types were clearly identifiable (e.g., *Wt1+* mesothelium, vascular smooth muscle, etc.), many of these clusters represented previously undefined groups of mesenchymal cells within the developing pancreas. This transcriptional analysis also identified candidate markers for these novel groups, which can identify distinct stromal populations in vivo. By linking related transcriptional groups across development, the authors determined that mesenchymal populations undergo dramatic transcriptional changes over time, and that mesothelial cells may give rise to multiple mesenchymal subtypes [34]. While this research suggests lineage relationships and unique functions between these different mesenchymal subpopulations, this concept has yet to be directly tested. Fortunately, the existence of Cre lines (*Wt1-GFPCre* and *Wt1-CreERT2* [63] targeting the mesothelium provides an opportunity to investigate the fate of this lineage during pancreas development. In the future, generation of Cre lines for other mesenchymal populations might allow functional characterization of other lineages throughout pancreas development. 

The relationship between mesenchymal heterogeneity in the embryo and the adult organ is still poorly understood. Many recently identified mesenchymal subpopulations have not been followed into adulthood, and as a result it is unknown which lineages persist in the adult organ, and how their functional role may change. Work by Landsman et al., Harari et al., and Sasson et al. demonstrates how genetic lineage tracing tools can be used to determine the contribution of different mesenchymal populations to adult tissues, and evaluate the function of these cells at developmental and postnatal timepoints [42,60,61].

## 3. Fibroblast Heterogeneity in the Healthy Pancreas

The bulk of the healthy pancreas is comprised of epithelial tissue, predominantly lobular acinar cells which excrete digestive enzymes into the small intestine via a ductal system. Mesenchymal cells, conversely, occupy less than 10% of the mature organ [64]. Despite their relatively low abundance, pancreatic fibroblasts are surprisingly diverse. Recent data has started to define these different populations in new levels of detail and suggest novel functions in the healthy pancreas.

One group of fibroblasts with a relatively well-defined origin and adult function are pancreatic pericytes. Derived (at least in part) from the *Nkx3.2*-expressing embryonic mesenchyme [60], these cells are found adjacent to endothelial cells throughout the adult pancreas [65]. Pericytes also express *NG*2, which can be used in combination with other markers and tissue analysis to identify pericytes in vivo [66,67]. Despite their presence throughout the pancreas, pericytes have primarily been studied in association with pancreatic islets. Ablating the *Nkx3.2* lineage of pericytes via DT led to glucose intolerance in adult mice [61]. Further, live imaging of pancreatic slice cultures suggests that pericytes respond to neural and islet-derived signals to regulate blood flow to the islets [68]. While these findings suggest that pericytes act as an essential mediator for the endocrine pancreas, their impact on the adult exocrine pancreas in still remains poorly understood. Existing genetic tools to target pericytes in vivo (e.g., *Nkx3.2-Cre*, *NG2-Cre*: [42,69]), could be exploited to determine how this cell type impacts other pancreatic functions. 

In contrast to the endothelial association of pericytes, mesenchymal stem cells (MSCs, also known as mesenchymal stromal cells) have been found in close association with pancreatic exocrine tissue [70,71]. MSCs are defined based on their ability to differentiate in vitro to adipocytes, chondrocytes, and osteoblasts [70,71,72,73]. MSCs can be identified through flow cytometry as CD45^−^;CD44^+^;CD49a^+^;CD73^+^;CD90^+^ [72,73]. It is unknown whether MSCs contribute to multiple mesenchymal cell types in the healthy pancreas, or whether they impact exocrine or endocrine function during homeostasis. In the absence of direct lineage tracing tools, isolated MSCs could be compared to other fibroblast populations through RNA sequencing analysis to determine: (1) whether MSCs have a unique interactome that may impact other cell types, and (2) whether MSCs share a transcriptional “lineage” with other mesenchymal populations [34], suggestive of MSC differentiation in the pancreas. 

In addition to MSCs, the exocrine portion of the pancreas is also the reported niche for pancreatic stellate cells (PSCs). PSCs have received substantial attention for their suggested contribution to fibrosis in the context of pancreatic disease [74,75,76,77,78]. PSCs have been defined by the presence of lipid droplets and their ability to “activate” α-Smooth Muscle Actin (αSMA) expression and deposit extracellular matrix (ECM) when isolated in 2D culture [64,79]. Additional PSC markers have been suggested that can be identified through staining, including Desmin and glial fibrillary acidic protein (GFAP) [64]. It is worth noting, however, that some of the markers frequently used to identify PSCs historically have either been non-specific (such as the neuron-detecting GFAP) or overlap with general fibroblast markers (e.g., Desmin) [80]. Further, no direct lineage tracing of this specific population in vivo has been done, making their developmental origin as well as their direct contribution to pancreatic fibrosis unclear. It is therefore worth considering whether the cells we refer to as “PSCs” are not a single cell type, but rather a heterogeneous group of different fibroblast cell types that are independently capable of contributing to pancreatic fibrosis.

Much like the development field, researchers studying heterogeneity within adult pancreatic fibroblasts have utilized large-scale transcriptome analysis to describe these populations on a molecular level. One research group performed single cell RNA sequencing on a combination of healthy adult mouse and human samples to identify transcriptional patterns consistent between species [35]. The researchers identified two distinct clusters of PSCs in both mice and humans, which they distinguished as “activated” (enriched for ECM-associated genes, including *COL1A1* and *FN1*) and “quiescent” (enriched for adipogenic genes, including *ADIRF* and *FABP4*) PSCs. Within the human samples, the researchers also found a subgroup within the “activated” PSCs that was enriched for cytokines, suggesting a subpopulation capable of modulating the immune cells in the healthy organ [35]. Although these data identify groups of PSCs with unique transcriptional profiles, it still remains unclear whether these groups have independent functions in the healthy pancreas. Further, the authors do not differentiate the “PSCs” analyzed from other fibroblasts, such as pericytes or MSCs. It is therefore unknown whether these subcategories are unique to PSCs or shared by multiple fibroblast populations within the adult pancreas. 

## 4. Fibroblast Heterogeneity in Pancreatic Injury and Disease

In many pancreatic diseases, fibroblast populations undergo dramatic remodeling. Chronic pancreatic injury activates fibroblasts, leading to the accumulation of dense, ECM-rich stroma [81]. This fibrosis not only disrupts healthy tissue function, but also increases the likelihood of developing pancreatic cancer [82]. One such disease, chronic pancreatitis, is strongly associated with increased morbidity and mortality, and severely reduces quality of life [83,84]. Unfortunately, this painful disease has limited treatment options [85]. Histologically, chronic pancreatitis is identified by a loss of mature acinar tissue, increased inflammation, and an abundance of fibrotic tissue [86]. Multiple rodent models have been generated in order to recapitulate this chronic human disease. These models utilize repeated exposure to exogenous compounds (e.g., Trinitrobenzene sulfonic acid [TNBS], caerulein, L-arginine, etc.), surgical intervention (e.g., pancreatic duct ligation) or a combination of both to induce pancreatic damage, acinar necrosis, and eventually fibrosis (for review see [87]). Notably, some of these compounds (e.g., caerulein) can be administered over a short period of time to induce acute pancreatitis, allowing researchers to study tissue recovery from acute pancreatic injury. Beyond these intervention-based models, recent work has produced genetic models of chronic pancreatitis [88,89,90,91]. These models display progressive pancreatic disease, either as a consequence of aging, or following an initial injury, and eventually develop the fibrosis typical of chronic pancreatitis. Further, several of these models utilize mutations analogous to those found in hereditary pancreatitis [88,90,91]. This provides an opportunity to study inherited risk factors of chronic pancreatitis, an avenue previously not possible with intervention-based models alone. 

Early studies of fibroblasts in chronic pancreatitis stained for stromal markers in a combination of patient samples and TNBS-treated rats, and found that PSCs were abundant in ECM-rich fibrotic areas [92]. In vitro studies further indicated that isolated PSCs could activate in response to a number of pancreatitis-associated cues, including transforming growth factor Beta (TGFβ), tumor necrosis factor alpha (TNFα), and ethanol [75,76,93,94]. More recent data has indicated that fibroblasts may affect chronic pancreatitis development through interactions with the immune system. In a caerulein-driven mouse model of chronic pancreatitis, researchers found that isolated PSCs alternatively activate macrophages via IL-4 and IL-13, and disrupting this interaction reduced fibrosis in vivo [95]. A different group utilized a combined caerulein and pancreatic duct ligation model, and found that PSCs are activated in response to immune complement signal C5a [96]. Genetic or pharmacological disruption of this interaction minimized the accumulation of fibrotic stroma [96]. Although these studies indicate this fibro-inflammatory cross-talk can promote pancreatitis, recent evidence has also suggested that these two populations communicate during tissue recovery. Disrupting stromal HH signaling in mice recovering from pancreatitis, specifically through the loss of a single *Gli1* allele, altered cytokine production in pancreatic fibroblasts, leading to a delay in tissue recovery [97]. Loss of GLI1 function has also been linked to inflammatory bowel diseases, suggesting that this HH-mediated immune modulation may be a common feature of chronic inflammatory diseases [98]. 

Although these studies have enhanced our understanding of fibroblasts in pancreatic disease, they have predominantly focused on PSCs. Given the evidence for distinct groups of fibroblasts in both development and the healthy adult pancreas, one current area of interest is how different populations of fibroblasts contribute to chronic pancreatitis. In a recent study, researchers collected human chronic pancreatitis samples and analyzed the expression of general fibroblast markers (e.g., αSMA, Desmin) as well as proposed stellate cell markers (e.g., CD34, NGFR, Tenascin C) through immunohistochemistry [99]. Interestingly, several of the proposed stellate cell subsets occupied distinct niches (e.g., periacinar vs. periductal) within the fibrotic microenvironment. Although this non-uniform expression is suggestive of fibroblast heterogeneity, unbiased single cell RNA sequencing efforts would provide better resolution into the stromal diversity of chronic pancreatitis. Further, lineage labeling of healthy fibroblasts (described above) prior to inducing chronic pancreatitis would directly determine the relative contribution of different fibroblast cell types to fibrosis. Such genetic tools could also be paired with DT-mediated modes of targeted ablation [61], allowing researchers to directly determine the functional contribution of these different populations to fibrosis development and maintenance.

## 5. Fibroblast Heterogeneity in Pancreatic Cancer

PDA is believed to arise from precursor lesions, most frequently pancreatic intraepithelial neoplasias (PanINs) [100]. The *KRAS* gene is mutated in the vast majority of low-grade PanINs [101,102], as well as in advanced disease [16,17,18]. Evidence from genetically engineered mouse models supports the notion that oncogenic KRAS is a driver of pancreatic cancer [19,20]. Fibroinflammatory stroma accumulates in concert with changes in the epithelium starting during the onset of transformation. In mice, acinar cells are the more common cell of origin for pancreatic cancer [103], although, under specific circumstances, ductal cells can be transformed as well [104,105,106]. Acinar cells dedifferentiate to a duct-like, progenitor-like cell type upon expression of oncogenic *Kras*, a process known as acinar-ductal metaplasia (ADM), which can be accelerated by pancreatic inflammation or induction of acute pancreatitis (for review see [23]). As acinar cells undergo ADM, fibroblasts become activated and express αSMA, as well as chemokines, growth factors, and extracellular matrix components. This fibroblast activation is dependent on oncogenic KRAS within the epithelium, and is reversed upon inactivation of oncogenic KRAS in a mouse model allowing inducible, reversible expression of oncogenic *Kras* in the pancreas [107]. In addition to mutant *Kras*, other oncogenic signals play a key role in driving fibrosis in PDA. Oncogenic *Kras* can coordinate with aberrantly active MYC to drive fibroblast expansion [108]. Once established, CAFs appear to be able to reciprocally support MYC expression via secretion of FGF1 [109], in a positive feedback mechanism that facilitates tumor growth.

CAFs were originally viewed as a uniform population. However, this notion has drastically changed, as a growing body of research across cancer types supports the idea that CAFs are heterogenous [110], both in their phenotype and their function within the TME (summarized in Table 1). In PDA, this heterogeneity exists across multiple axes: physical location within the TME, ability to respond to different intercellular signals, and transcriptional profile.

### 5.1. Positional Heterogeneity

Evidence of CAF heterogeneity in PDA began to accumulate with descriptive reports of nonuniform staining of fibroblast markers. In human and mouse PDA, researchers found that common fibroblast markers such as αSMA, podoplanin, platelet-derived growth factor receptor Alpha/Beta (PDGFRα/β), fibroblast specific protein 1 (FSP1), fibroblast activating protein (FAP), and desmin varied in their staining intensity, distribution, and overlap throughout the tumor tissue [6,26,99,111,116,117,118,119]. In particular, fibroblasts immediately adjacent to tumor cells appeared distinct from their counterparts at distal locations. Recent studies have sought to characterize these spatially distinct populations. In both KPC mice and human PDA, tumor-adjacent stroma expressed higher levels of αSMA, while more distant fibroblasts expressed IL6 [111,112]. These populations were named “myCAF”, for myofibroblasts, given the high level of αSMA expression, and “iCAF”, characterized by higher expression of inflammatory cytokines. Further, different signals appeared to drive the differentiation of each type of fibroblasts, with TGF-β promoting myCAFs and IL1/Jak-stat driving iCAFs [111,112]. Although these recent studies have enhanced our understanding of spatially distinct fibroblast populations, the field still lacks a comprehensive spatial “map” of all of the different fibroblast populations in PDA. Techniques such as multiplex immunohistochemistry (mIHC) could be utilized to determine the spatial distribution of these newly identified fibroblast populations in relation to other elements of the TME [120]. Determining how this fibroblast “map” varies across PDA subtypes and disease stages would elucidate the niches of these different fibroblast sub populations.

### 5.2. Interactive Heterogeneity

Throughout the process of pancreatic cancer progression, the developing tumor cells produce a vast array of secreted proteins that activate signaling within the TME (Figure 1). The exact response to these different signals varies between different CAF populations, and the exact role of these signaling interactions has been the source of significant controversy. Although many signaling pathways have been identified in the stroma of PDA, the roles of Hedgehog and TGF- β signaling have been particularly controversial. 

#### 5.2.1. Hedgehog Signaling

HH signaling as a regulator of pancreatic cancer was first described over a decade ago, when SHH ligand was found to be expressed in a majority of pancreatic tumor cell lines and human patients [17,121,122]. Initially described as upregulated in cancer cells, HH signaling was later shown to function exclusively in a paracrine manner. PDA tumor cells secrete HH ligands, mainly SHH and indian hedgehog (IHH), which bind to the transmembrane receptor Patched 1 (PTCH1) on fibroblasts [123,124,125]. Binding of HH ligand to PTCH1 releases the inhibition of smoothened (SMO), which in turn activates the GLI family of HH transcription factors (for review see [126]). Despite multiple studies investigating HH signaling mechanisms in PDA, the functional role of HH signaling in pancreatic cancer is still not clear. Disrupting the HH pathway by pharmacological SMO inhibition slowed tumor growth in a subcutaneous tumor model [123], and slightly prolonged survival when combined with chemotherapy in a tumor-bearing KPC mice [9]. These data support a model in which HH signaling promotes tumor growth, and could potentially be targeted to improve patient survival. This led to a clinical trial of the small molecular SMO inhibitor, IPI-926 (Infinity Pharmaceuticals), and parallel trials of a separate SMO inhibitor, GDC-0449 (vismodegib, Genentech) in patients with pancreatic cancer. Unfortunately, the Infinity Pharmaceuticals trial had to be terminated early due to decreased survival in the IPI-926 experimental group [127], and the GDC-0449 trials provided little to no benefit to patients [128,129].

After the clinical trial results became public, the role of HH signaling in pancreatic cancer was revisited in two studies that inactivated *Shh* expression in mouse models of pancreatic cancer. In both studies, investigators observed that loss of *Shh* led to faster development of invasive tumors and shortened survival [13,14]. A caveat of both studies is that SHH is only one of three HH ligands, and, as mentioned before, at least two (*Shh* and *Ihh*) are expressed in pancreatic cancer [13,123]. Thus, it is likely that inactivation of SHH led to a reduction, but not ablation of HH pathway activity. HH signaling has profoundly different effects in a dosage-dependent manner in embryonic development (for review see [130]). To address the possible effects of different HH signaling levels in PDA, our group inactivated the HH coreceptors GAS1, BOC, and CDON in pancreatic fibroblasts [12]. GAS1, BOC, and CDON are classically thought to promote HH signaling [131,132,133,134], and genetic deletion of two or three co-receptors in pancreatic fibroblasts reduced or completely inhibited HH signaling levels, respectively [12]. Interestingly, fibroblasts with reduced (but not ablated) levels of HH signaling promoted tumor growth, while HH-unresponsive fibroblasts did not [12]. It will therefore be important to consider how levels of HH-response vary throughout the TME, and how these different HH-responding populations are influencing pancreatic tumor development.

Beyond the roles of canonical HH signaling in PDA, recent data has revealed that HH pathway components can have non-canonical functions during pancreatic cancer progression. Conditional deletion of *Smo* in the stroma led to a higher prevalence of ADM, and promoted cancer cell growth [135,136]. Although *Smo* null fibroblasts were no longer able to respond to HH signaling, they surprisingly upregulated *Gli2* expression. This compensatory upregulation of *Gli2* led to the production of TGFα, which in turn promoted metaplasia [135]. These data could partially explain the accelerated PDA progression observed in HH antagonized mouse models [13,14]. Thus, it is worth exploring how different populations of pancreatic fibroblasts respond to HH signaling in vivo, and whether the activity of HH pathway components varies throughout the TME. 

#### 5.2.2. TGF-β Signaling

Another signaling pathway that is highly active within the TME is the TGF-β pathway. Some of the first studies into pancreatic fibroblasts identified TGF-β signaling as a pro-fibrotic pathway [75,79]. More recently, TGF-β has been shown to maintain a myofibroblast “myCAF” subpopulation in pancreatic cancer (described in detail below, [111,112]). However, dissecting the TGF-β interaction network is complex, as multiple cell types (including fibroblasts) within the TME express TGF-β ligands, allowing for both autocrine and paracrine modes of signaling [76,78,137,138]. Further, although TGF-β activity is typically thought to limit cell cycle progression in healthy tissue and therefore act as a tumor suppressor, the pathway is aberrantly activated in advanced pancreatic disease [139]. There have been conflicting reports over whether disruption of TGF-β signaling is protective [140,141] or deleterious [142,143,144,145]. It is worth noting that these studies utilized different models in order to disrupt TGF-β signaling. The studies reporting that loss of TGF-β can promote progression utilized genetic loss of TGFBR2 [142] or conditional deletion of *Smad4*, a downstream regulator of TGF-β signaling [143,144,145], in *Ptf1a-* and *Pdx1-Cre* driven *Kras^G12D^* models. On the other hand, studies that described protective effects from loss of TGF-β signaling utilized *Ela-KRAS^G12D^* transgenic mice with global loss of 1 copy of TGFBR1 [140,141]. Therefore, the exact model used and the mode of TGF-β disruption may have a significant impact on disease phenotype. 

Importantly, the role of TGF-β also varies between cell types within the TME [141,146,147]. Targeting TGF-β signaling specifically in CD8+ T-cells slowed PanIN progression [148]. Conversely, myCAFs rely on TGF-β to maintain their tumor-restraining function [112], and removing a key source of TGF-β (via regulatory T cell depletion) leads to myCAF reprogramming, increased immunosuppression, and accelerated neoplastic progression [138]. Within the tumor epithelium, aberrant TGF-β signaling has been linked to epithelial-mesenchymal transition (EMT), promotion of invasion, chemoresistance, and -although the concept is controversial- cancer cell stemness [149,150,151,152,153]. The exact effect of TGF-β in PDA depends on Smad4 status. Tumor cells that express wildtype Smad4 undergo EMT following TGF-β stimulation, but ultimately apoptose [154]. In contrast, mutations in Smad4 synergize with other oncogenic mutations (e.g., *Kras^G12D^*) to drive PDA progression and tumor growth [143,154,155,156]. These data highlight the complex cross talk between the immune cells, tumor cells, and CAFs, and the importance of detailed characterization of these different signaling pathways within the TME.

### 5.3. Transcriptional Heterogeneity

The recent explosion of single cell sequencing technology has brought insight into the heterogeneity of fibroblasts in pancreatic cancer. A series of papers classified pancreatic CAFs from a KPC mouse model and human PDA into replicable subsets with distinct transcriptional profiles and activation pathways. Pancreatic CAFs have been generally characterized by either a myofibroblast “myCAF” or inflammatory “iCAF” phenotype [37,111,112]. The myCAF population is directly adjacent to tumor cells, expresses high levels of αSMA, and is maintained by the TGFβ pathway [111,112]. These αSMA-expressing myCAFs are believed to be tumor restricting, consistent with earlier studies that depleted αSMA-expressing fibroblasts and observed promotion of PDA progression [15]. A separate population, the distal iCAF population, expresses IL6 among other inflammatory chemokines and cytokines, and relies on Il1/Jak-Stat signaling [111,112]. These populations are conserved across PDA models, as single cell RNA sequencing analyses from the *p16-Ink4a/p19-Arf* loss of function KPP model also identified an IL1-driven iCAF-like population and a TGFβ -driven myCAF-like population [38]. Another recent single cell RNA-seq study analyzed the fibroblasts throughout PDA progression, and determined that myCAFs and iCAFs may originate from distinct populations in the healthy and early lesion stages [36]. A third antigen-presenting “apCAF” population has recently been described in both KPC mice as well as humans [37]. However, this latter antigen-presenting population might represent the mesothelium in pancreatic cancer [38]. Although these different studies all identified similar subtypes of fibroblasts, the exact profile of the fibroblast groups varied depending on the specific genetic mouse model used [36,37,38]. These results highlight the importance of stromal—epithelial cross talk in the TME and the importance of model selection (for review see [157]).

Notably, these CAF subset designations appear to be interconvertible in vitro [112] and share a common base fibroblast program [38,112]. For example, inflammatory CAFs can also contribute to ECM deposition by expressing hyaluronan and collagens [37,38]. Thus, the boundaries between CAF different subtypes are likely fluid and context-dependent. It is therefore worth tracing these different subtypes in vivo throughout different stages of PDA progression, to see how different CAF populations respond to a changing TME. 

## 6. Origin of Cancer-Associated Fibroblasts

Pancreatic CAFs have long been assumed to derive from a resident population of PSCs. However, growing data support the possibility that multiple fibroblast populations in the healthy pancreas may contribute to the heterogeneity of CAFs [36]. Lineage-tracing experiments in fibrosis models in other organs such as the liver, kidney, heart, skin, and lung, consistently demonstrate that various resident fibroblast populations proliferate in response to injury to contribute to fibrosis [158,159,160,161,162,163,164]. Beyond PSCs, MSCs have been identified in healthy tissue and are expanded in neoplastic tissue, raising the possibility that MSCs can also contribute to neoplastic stroma [72,73]. Further, it has been suggested that some CAFs may arise in part from bone marrow-derived cells [165], pericytes, or endothelial cells (for review see [166]). Single-cell sequencing data have also suggested that some stromal cells arise in part from cancer cells that have transformed through EMT, though they account for a relatively small portion of the stroma [38]. Although these populations have been suggested to give rise to CAFs, researchers have been largely been limited by a lack of effective lineage-tracing tools for these populations in vivo. It is therefore difficult to conclusively determine the relative contributions of these different populations to the PDA stroma

Our group has investigated the origin of stromal fibroblasts by lineage tracing two largely distinct populations of fibroblasts present in the normal pancreas, characterized by expression of *Gli1* (a downstream effector of HH signaling) and *Hoxb6* (a homeobox gene with a known role in the embryonic pancreatic mesenchyme [56]). We found that *Gli1*-expressing fibroblasts, localized in the perivascular region of the healthy pancreas, expand during carcinogenesis, giving rise to about half of the total stromal fibroblasts. In contrast, *Hoxb6*-expressing cells do not expand during carcinogenesis, and are sparse in neoplastic lesions [28]. While *Gli1*+ fibroblasts largely give rise to myCAFs, it does not appear that they exclusively contribute to this population. Further, the functional role of *Gli1* in this process and the role of HH signaling in the formation of myCAFs both remain unclear. Future work is required to determine the origin of the remaining stromal fibroblasts. 

Beyond genetic lineage tracing, recent studies have utilized computational methods to investigate the origin of CAFs. Researchers have used gene ontogeny analysis to identify related transcriptional programs between healthy and diseased tissue throughout lesion development, and infer the origin of pancreatic CAFs through pseudotime [38]. This analysis linked myCAFs and iCAFs to two distinct groups in the healthy pancreas, suggesting that these different CAFs originate from different fibroblast populations within the healthy organ [38]. A separate study performed single-cell RNA sequencing across multiple PDA mouse models throughout PDA progression, and used hierarchical clustering to compare fibroblast populations from healthy to advanced PDA. Consistent with the study described above, this group also identified two distinct populations in the healthy pancreas that remained distinguishable throughout PDA progression [36]. Further work has suggested that myCAFs and iCAFs may emerge at different stages of PDA progression. While the myCAF phenotype was detected throughout PDA progression, the iCAF phenotype was only observed in PDA [113]. These studies indicate that myCAFs and iCAFs evolve from unique fibroblast populations in the healthy tissue. However, since these populations were linked in silico, this dataset can only suggest, but not prove, a lineage relationship between healthy resident populations and CAFs. 

## 7. Challenges in Studying Fibroblast Heterogeneity

Although the research described above has provided novel insight into fibroblast heterogeneity, the field is currently challenged by conflicting data. It has been particularly difficult to come to a consensus regarding the specificity of fibroblast population markers and their expression patterns. For example, commonly used fibroblast markers such as PDGFRα, αSMA, PDPN have each been described as a pan-stromal marker by some groups, and a subtype-specific marker by others [37,38,111,167,168]. Such inconsistencies in the field have made it difficult to generate a unified model of stromal heterogeneity. As such, a number of considerations must be accounted for when evaluating fibroblast populations in PDA. 

The first consideration is the source of the tissue. Although human patients and mouse models have many similarities within the TME [37,38,169], the progression of the disease differs fundamentally between these two groups; the former typically acquiring mutations and experiencing histological changes over the course of years, while the latter experiences accelerated disease progression driven by oncogenic mutations as early as embryonic development. Obtaining data from human patients is also complicated by natural variability in patient cohorts (age, sex, lifestyle, stage of disease, etc.), as well as the relative scarcity of tissue. Data from mouse models can also be difficult to compare, considering the multitude of models (e.g., KPC, KIC/KPP) utilizing different oncogenes to drive carcinogenesis. A recent analysis of different mutant p53 alleles (expressing a dominant *R172H* mutation versus p53 loss-of-function) in KPC models observed key differences in the stiffness of the ECM matrix and susceptibility to chemotherapy [170], which suggests the slight genetic variations of modeling can impact the stromal TME. In addition, using an in vitro cell system introduces environmental cues, either by tumor conditioned media [168], signaling molecules [112], or dimensionality of the culture conditions (2D vs. 3D) [37,170] that impact the cell state and phenotype. Since CAF subtypes may interconvert with one another given the appropriate cue [112], care must be made to not overgeneralize findings.

Another major source of data discrepancy may be due to technical variability while obtaining and processing samples. Different studies have utilized very different protocols to isolate, digest, sort, sequence, and analyze stromal populations, and as a result there are dramatic differences in cell types captured and abundance of fibroblasts. This can have very real impact on sequencing efforts, as low or biased yields will limit the ability to fully capture all of the fibroblast populations within the TME. This highlights a critical need in the field to develop optimized protocols to enrich for stromal populations. More effective stromal isolation is crucial to fully capture the fibroblast diversity that exists within the TME.

## 8. Targeting Cancer-Associated Fibroblasts in the Clinic

While our understanding of CAFs in PDA is still evolving, interactions between CAFs and the TME have already been leveraged for potential therapies. CAFs have many reported tumor-promoting functions, including metabolic support, recruitment of immunosuppressive cells, and creating a physical barrier to drugs through the ECM (for review see [7,157,171]). Many clinical trials have sought to improve patient survival by targeting these CAF-associated elements of the TME. However, recent data demonstrates that CAFs also have tumor-restricting roles [12,13,14,15], making the clinical targeting of the stroma all the more complex. 

The concept of targeting pro-tumor intercellular signaling pathways has been explored therapeutically with mixed results. A notable attempt to target CAF signaling therapeutically involved inhibiting the HH signaling pathway. Although the HH pathway inhibitor IPI-926, when delivered with the chemotherapeutic gemcitabine, was reported to improve survival in mouse models of PDA [9], neither IPI-926 nor GDC-0449 (Vismodegib) significantly improved survival in human patients (described in detail above, [128,129,172]). In addition to HH, vascular endothelial growth factor (VEGF) signaling has been targeted as a potential anti-stromal therapy. VEGF is produced by both tumor cells as well as stromal cells [173,174], and VEGF expression correlates with decreased survival in human patients [175]. Clinical trials tested anti-VEGFa monoclonal antibodies (bevacizumab) and small-molecule inhibitors of VEGF receptors (Axitinib) in combination with chemotherapy, but neither strategy improved survival in human patients [176,177]. Although it is difficult to pinpoint the exact reasons these trials failed, one underappreciated factor could be the heterogeneity of the stroma, in which distinct populations respond differently to treatment. 

Researchers have also tried to target the contribution of CAFs to chemoresistance. Numerous studies have shown that CAFs can directly protect tumor cells from common therapies, including radiation and chemotherapy [8,178,179,180,181,182]. Beyond these direct effects, CAF-derived ECM has been shown to support PDA through multiple mechanisms. A potential role of CAFs is to create a physical barrier, through accumulation of dense ECM which in turn drives high interstitial pressure [183,184]. This increase in interstitial pressure has been shown to restrict blood flow and impair drug delivery to the site of the tumor [9,10,11]. As a result, the pancreatic cancer TME is nutrient poor and hypoxic, creating an environment that is immunosuppressive and drives metabolic reprograming and chemoresistance in cancer cells [185,186]. Beyond these physical effects, the stroma-derived ECM can also act as a nutrient source for tumor cells [187]. A recent study demonstrated that this process relies (in part) on *NetG1* expression in CAFs, as loss of stromal *NetG1* led to reduced tumor cell survival in response to nutrient deprivation [188]. The array of tumor-supporting roles has made the ECM an attractive target for new therapies.

One particular component of the ECM, hyaluronic acid (HA), is frequently overexpressed in PDA patients [10,11], and can promote tumor growth and facilitate drug resistance [189]. Depleting HA with a stabilized version of hyaluronidase (PEGPH2O) in KPC mice decreased interstitial pressure, increased drug delivery to the site of the tumor, and improved survival in combination with gemcitabine [10,11]. A phase II clinical trial found a minor improvement (1 month) in progression-free survival for all patients receiving PEGPH2O alongside dual chemotherapy (Nab-paclitaxel and Gemcitabine), and a moderate benefit (4 months) for patients with high HA expression [190]. This suggested that this might be a viable strategy specifically for HA-high patients. However, a phase III clinical trial that specifically enrolled HA-high PDA patients failed to significantly improve overall survival [191,192]. An alternative strategy has utilized angiotensin receptor blockers (losartan) to target HA in combination with other ECM components, specifically collagen I [193]. Losartan reduced *Col1* and *Has1-3* expression in CAFs, leading to a decrease in collagen and HA deposition in orthotopic tumor models. This reduction in ECM led to improved tumor perfusion, and improved survival when combined with chemotherapy [193]. Although these pre-clinical data are encouraging, further long-term studies in spontaneous tumor models will be needed to determine if this is a potential viable strategy in the clinic. 

While many of the strategies described above antagonize the pro-tumorigenic products of CAFs, some researchers have sought to reprogram CAFs into a less protumorigenic, quiescent state. Both RA (ATRA) and vitamin D analogs (calcipotriol) have been used to reverse the activated state of CAFs, leading to broad transcriptional changes consistent with a shift towards a quiescent cell identity [194,195]. KPC mice treated with these components displayed a decrease in αSMA, a reduction in tumor cell growth, and an increase in cell death. Further, combination therapy of calcipotriol with gemcitabine improved survival in KPC mice [195]. However, a recent study reported that calcipotriol can have an immunosuppressive effect on CD8+ T cells [196], which may limit the long-term efficacy of vitamin D analogs in PDA patients, and might explain the generally disappointing clinical trials.

Another therapeutic avenue is to target the immunosuppressive activities of CAFs. Immunotherapies that utilize checkpoint inhibitors have shown minimal efficacy against pancreatic cancer, and one proposed explanation is that CAFs interfere with an effective anti-tumor immune response [197]. Therefore, targeting the immunosuppressive functions of CAFs or depleting immunosuppressive CAF populations might make immunotherapies more effective. The *Fap*-expressing population of fibroblasts has been identified as a potential population to target, as depleting *Fap*-expressing cells via a transgenic DTR expression or transferring FAP-targeted chimeric antigen receptor T-cells slowed tumor growth in both subcutaneous and KPC models [5,6,115]. Depleting FAP+ fibroblasts in immunodeficient mice did not impact tumor growth, suggesting that the tumor-promoting role of the FAP+ CAFs is not due to direct impacts on tumor cells, but rather through manipulations of the immune response [5,6]. However, depletion of FAP+ fibroblasts systemically via DT led to a loss in muscle mass and hematopoeitc cells, indicating that FAP+ depletion is not a tenable therapeutic option [114]. Fortunately, FAP+ cells secrete CXCL12, a cytokine capable of restricting CD8+ T-cell infiltration via the targetable receptor CXCR4. Pharmacologic CXCR4 inhibition effectively reduced tumor growth, and was especially effective when paired with an immune checkpoint inhibitor α-PD-L1 [6]. Although these results are promising, the efficacy of these treatment strategies at prolonging survival in PDA is still unknown. A clinical trial investigating the combined efficacy of CXCR4 and PD-1 inhibitors in pancreatic cancer is scheduled to begin in the coming months (NCT04177810).

A potential complication with the strategies described above could be that these inhibitors may be affecting the stroma too broadly, resulting in the disruption of tumor-restricting fibroblast populations. Thus, it is worth considering whether more targeted approaches against specific pro-tumor CAF populations would be a more effective therapy options Recent work (summarized above) has proposed at least three CAF populations in PDA: myCAFs, iCAFs, and apCAFs [37,111,112]. Of these three, iCAFs in particular have been shown to have pro-tumor activity via IL1/JAK/STAT signaling [112]. Inhibition of JAK-STAT signaling reduced the expression of inflammatory cytokines in PSCs and reduced tumor growth in KPC mice [112]. A phase I clinical trial (NCT02550327) will be testing the efficacy IL1-receptor antagonist (Anakinra) in PDA, which may restrict the pro-tumor activity of iCAFs. Although this trial acknowledges the heterogeneity of CAF populations, several concerns still remain. Although JAK-STAT inhibition reduced tumor growth over the course of 10 days [112], it is unclear how prolonged treatment would affect the survival of KPC mice. It is also unclear whether this timeline would effectively suppress the iCAF phenotype over extended periods of time, or whether other CAF populations would change to fill the functional niche of iCAFs. More fundamentally, the in vivo role of these different populations throughout pancreatic cancer progression is still poorly understood, as most of the inferred function so far has been based on in vitro experiments or correlative data following systemic drug treatments. In the absence of targeted in vivo manipulations of these different populations, it is difficult to fully understand how these different populations interact with each other and within the dynamic TME. Fortunately, single cell sequencing efforts by the same group have identified a suite of potential markers that correspond to iCAFs (e.g., *Clec3B*, *Ly6C*, *Col14a1*), myCAFs (e.g., *Acta2*, *Thy1*, *Col12a1*), and apCAFs (e.g., *H2-Ab1*, *Saa3*, *Slpi*) [37], opening the possibility for CAF-subtype genetic tools to be developed. Inducible systems (e.g., CreERT2) targeting a subset of fibroblasts in a temporally defined manner would avoid disrupting other populations expressing these same markers during embryonic development, and could be paired with FlpO-based (*Ptf1a-FlpO*, *Pdx1-FlpO*) modes of pancreatic carcinogenesis (i.e., KF, KPF, [25,26,27,28]) to manipulate of CAF populations throughout PDA progression in vivo. Without closely characterizing the dynamic stromal responses to proposed anti-CAF treatments, future clinical trials will likely face the same disappointing results.

## 9. Conclusions and Future Directions

Fibroblasts are a dynamic and complex component of the pancreatic stroma, from development to adulthood to cancer. Advances in transcriptomic sequencing tools in both animal models and human patients have helped define distinct populations of fibroblasts, throughout many stages of development and disease. Many questions still remain, however, and several coordinated efforts are necessary in order to advance the study of fibroblast heterogeneity. First, the latest developments in large scale transcriptomics need to be paired with spatial information in vivo. Advances in mIHC have been used to characterize spatial relationships between T cell populations and cancer cells within the TME, and revealed that high cytotoxic T cells in close proximity of cancer cells correlated with improved survival [120,198]. Alongside these advances in imaging technology, researchers have recently developed computational tools that utilize deep learning to automate mIHC analysis in human PDA [199]. The marriage of detailed spatial analysis with transcriptional profiling [200] could be applied towards the study of stromal diversity to provide a more comprehensive view of cellular heterogeneity in PDA. Such analysis could also be used clinically, to provide a much more detailed view of an individual patient’s TME and tailor precision treatments accordingly. 

Further, the generation and utilization of stromal genetic lineage tracing in animal models will allow us to follow these diverse populations throughout different stages of pancreas development and disease. With the advances in dual recombination models [25,26,27,28] it is possible to identify and manipulate multiple populations within the in vivo microenvironment to investigate how these cell types evolve, function, and interact. Such models can also be adapted to deplete specific cell types of interest (via DTR expression) or conditionally delete stromal genes of interest (via floxed alleles) to ask more targeted questions in vivo. As we gain deeper insight into the complex nature of fibroblast heterogeneity, we will be better poised to leverage these populations in the clinic. 

## Figures and Tables

**Figure 1 cells-09-02464-f001:**
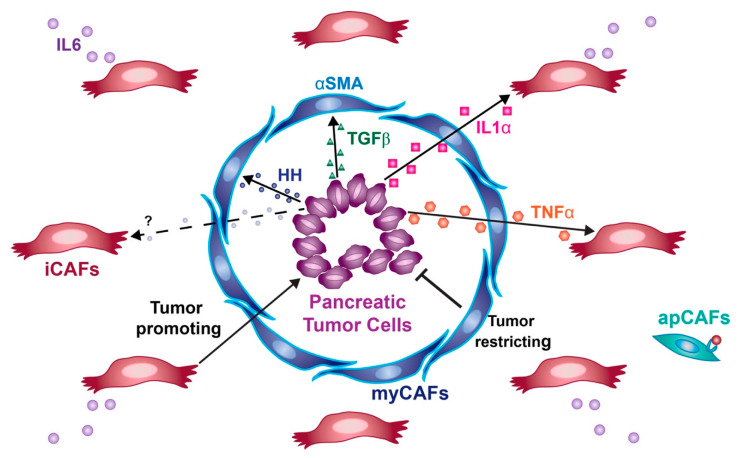
Key stromal signals in PDA. myCAFs (blue) respond to HH and TGF-β ligands secreted by tumor cells (purple), express high levels of αSMA, and restrict tumor growth. iCAFs (red) are maintained by tumor-derived IL1α and TNFα, express IL6, and promote tumor growth. apCAFs (teal) express MHC Class II and can present antigens to T cells.

**Table 1 cells-09-02464-t001:** Fibroblast Populations in Pancreatic Cancer.

Fibroblast Population	Defining Features	Known Function in PDA	Key References
myCAFs	*αSMA^high^* TGF-β dependent Tumor-adjacent	Tumor-restricting	[15,36,37,38,111,112,113]
iCAFs	*IL6+* IL1/Jak-stat dependent Distant from tumor	Tumor-promoting	[36,37,38,111,112,113]
apCAFs	MHC Class II+ Have also been described as mesothelial cells	Antigen presentation to T cells	[37,38]
FAP+ Fibroblasts	*Fap+*	Immunosuppressive	[5,6,114,115]
*Gli1* Lineage	Derived from *Gli1+* fibroblasts in healthy pancreas	Not fully understood Partially contributes to myCAFs	[28]
MSCs	CD45^−^;CD44^+^;CD49a^+^;CD73^+^; CD90^+^	Tumor-promoting	[72,73]

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
