# Peer review of "Pancreatic Fibroblast Heterogeneity: From Development to Cancer"

_cells, 2020, doi:10.3390/cells9112464_

Round 1
Reviewer 1 Report
This is a timely review on the topic of fibroblasts in pancreatic cancer. The review is well organized and provide interesting insights in the field.
Here some suggestion to improve the paper:
1) The authors are strongly encouraged to provide a summarizing figure regarding fibroblasts-heterogeneity in pancreatic cancer, representing schematic morphology and involved pathways.
2) The authors shoudl provide some parallelisms with the same/similar pathways involved directly in pancreatic cancer / neoplastic cells. For example, TGF-beta signalling and the epithelial-mesenchymal transition should be better discussed, also in the light of seminal papers in this field, which have highlighted (as only partly addressed in this review) the important role of these pathways in pancreatic cancer (important examples are PMID: 32429474; 31398893 ; 31331827 ; 30669452 ; 32661742 ; 30891677 ).
3) The authors shoudl provide a summarizing table presenting the major findings they reviewed. Both summarizing figure and table will be of great help to the reader
4) The authors should better address the topic of chemo-resistance related to extracellular matrix and fibroblast (and of other inflammatory cells) in a dedicated paragraph, which should be very exhaustive and comprehensive, since chemo-resistance represents to date one of the most important reasons for studying fibroblasts in pancreatic cancer.
5) in future development, the authors should mentioned and discuss the potentiality of artificial intelligence / digital pathology in studying these types of cells.
Reviewer 2 Report
This review from Garcia and Scales et al provides a well-written and largely comprehensive overview of our current knowledge about the heterogeneity and roles of fibroblasts in pancreatic cancer. I recommend its publication in Cells.
However, I have a few points that should be included in the manuscript.
- If in line with the journal guidelines, as it appears from other reviews, I encourage the authors to include some tables, schematics and figures that highlight/summarize the main concepts discussed.
- On line 71, the authors mention single-cell RNA-sequencing of some other cancer types. I would also include the analysis by Puram et al 2017 Cell in head and neck cancer.
- In section 4, the authors should briefly clarify that depending on the duration of the treatment, caerulein models and other models can induce acute, rather than chronic pancreatitis. They should also acknowledge recent genetically engineered mouse models of pancreatitis (e.g. Geisz et al 2018 Nat Comm; Gui et al 2019 JCI; Engle et al 2019 Science; Huang et al. 2019 Gastroenterology).
- In section 5, the authors acknowledge work showing that fibroblast activation is dependent on oncogenic KRAS. The author should also acknowledge work showing the role of MYC in fibroblast activation in pancreatic cancer (Sodir et al 2020 Cancer Discovery).
- In section 8, the authors should acknowledge other fibroblast-targeting strategies, such as calcipotriol (e.g. Sherman et al 2014 Cell), losartan (e.g. Chauhan et al 2013 Nat Comm) and ATRA (e.g. Froeling et al 2011 Gastroenterology).
